# Issues for Continual Learning in the Presence of Dataset Bias

**Donggyu Lee**[1*], **Sangwon Jung**[2*], **Taesup Moon** [2,3,4†]

[1] Department of Electrical and Computer Engineering, Sungkyunkwan University
[2] Department of Electrical and Computer Engineering, Seoul National University
[3] SNU-LG AI Research Center   [4] ASRI/INMC/IPAI/AIIS, Seoul National University
ldk308@skku.edu,{s.jung,tsmoon}@snu.ac.kr

## Abstract

While most continual learning algorithms have focused on tackling the stability-plasticity dilemma, they have overlooked the effects of the knowledge transfer when the dataset is *biased* — namely, when some unintended spurious correlations, not the true causal structures, of the tasks are learned from the biased dataset. In that case, how would they affect learning future tasks or the knowledge already learned from the past tasks? In this work, we design systematic experiments with a synthetic biased dataset and try to answer the above question from our empirical findings. Namely, we first show that standard continual learning methods that are unaware of dataset bias can transfer biases from one task to another, both forward and backward. In addition, we find that naively using existing debiasing methods after each continual learning step can lead to significant forgetting of past tasks and reduced overall continual learning performance. These findings highlight the need for a causality-aware design of continual learning algorithms to prevent both bias transfers and catastrophic forgetting.

## Introduction

Continual learning (CL) is essential for a system that needs to learn (potentially increasing number of) tasks from sequentially arriving data. The main challenge of CL is to overcome the *stability-plasticity* dilemma (Mermillod, Bugaiska, and Bonin 2013). Namely, when a CL model focuses too much on the stability, it would suffer from low plasticity for learning a new task (and vice versa). Recent deep neural networks (DNNs) based CL methods (Kirkpatrick et al. 2017; Jung et al. 2020; Li and Hoiem 2017) attempted to address the dilemma by devising mechanisms to attain stability while improving plasticity thanks to the *knowledge transferability* (Tan et al. 2018), which is one of standout properties of DNNs. Namely, while maintaining the learned knowledge, the performance on a new task (resp. past tasks) is improved by transferring of knowledge of past tasks (resp. a new task). Such phenomena are called the forward and backward transfer, respectively.

While such DNNs based approaches for CL have been successful to some extent, they have not explicitly considered a more realistic and challenging setting in which the *dataset bias* (Torralba and Efros 2011) exists; *i.e.*, a distribution of test dataset could be different from that of training dataset for each task. In such a case, it is widely known that DNNs often dramatically fail to generalize to the out-of-distribution test data due to learning some unintended spurious correlations, not the true causal relations (Sagawa et al. 2020; Bahng et al. 2020). For instance, a DNN that classifies birds in the sky perfectly may fail on classifying images in which birds are outside the typical sky background when the model has learned a *shortcut* strategy relying on the background (Geirhos et al. 2020). There have been many attempts to address this issue, with earlier approaches (Nam et al. 2020; Liu et al. 2021) often based on empirical findings about DNNs, resulting in suboptimal results. More recently, there have been efforts to address the bias issue in a more principled way by using a structural causal model (SCM) to clarify the causal relationship between input, label, bias, and context priors (Liu et al. 2022; Seo, Lee, and Han 2022). By obtaining direct causal effects from inputs without the confounding influence of bias, these approaches have shown improved performance on various vision tasks, including object classification (Liu et al. 2022), semantic segmentation (Zhang et al. 2020) and few-shot learning (Yue et al. 2020), highlighting the importance of causal learning in solving the bias problem.

Now, we claim that the issue of learning spurious correlations, not the true causal relations, in the context of CL can be a significant problem because it can lead to the transfer of bias from one task to another. In a recent study (Salman et al. 2022), it is shown that the transfer of bias can even occur when fine-tuning pre-trained models on downstream tasks. In CL, this issue can be potentially exacerbated since it involves learning a sequence of tasks, and the transferred bias can affect not only the future tasks, but also the past tasks. Additionally, the severity of bias transfer in CL may be greater depending on how the learned knowledge is utilized. However, to the best of our knowledge, there is a lack of research that is carefully investigating this issue for CL.

To that end, we show that when a certain task in a CL scenario contains a dataset bias, applying naive CL methods to learn such a task would be problematic since they can

---

*These authors contributed equally.
†Corresponding author.

maintain unwarranted knowledge (*e.g.*, background bias) and transfer it to future or past tasks. To test this, we construct a synthetic dataset with color bias, and systematically conduct extensive experiments on various two task scenarios with varying levels of bias. We quantitatively identify that the forward and backward transfer of bias indeed exist when naive CL methods are applied. More specifically, we show that a typical CL method preserves the knowledge such that the bias of the knowledge learned from the past task is reused to train on a new task (*i.e.*, forward transfer of bias), resulting in severer bias for the new task. Furthermore, it is shown that the biased knowledge learned from the current task also affects the decision rules for the past tasks to be biased (*i.e.*, backward transfer of bias), and a naive debiasing for the current task could also cause the catastrophic forgetting of the past task. Our results clearly call for a principled, novel approach for taking the causal learning into account while continual learning from potentially biased datasets, in order to prevent both bias transfers and forgetting.

## Case Studies of Bias Transfer in CL

### Experimental Settings

**Dataset.** We use Split CIFAR-100 (Zenke, Poole, and Ganguli 2017; Chaudhry et al. 2019; van de Ven, Siegelmann, and Tolias 2020), which divides CIFAR-100 into 10 tasks with 10 distinct classes. To study bias transfer, we modify Split CIFAR-100, such that half of the classes in each task are skewed toward the grayscale domain and the other half toward the color domain. Namely, given a skew-ratio $\alpha \geq 0.5$, the training images of each class are split into $\alpha$ and $1 - \alpha$ ratios for each domain. We set 6 bias levels by dividing the range from 0.5 to 0.99 evenly on a log scale for systematic control of the degree of bias.

**CL Scenario.** We consider a task-incremental learning scenario (Van de Ven and Tolias 2019) in which a task identifier $t \in \mathcal{T} \triangleq \{1, 2, 3, \cdots\}$ is given during inference time and further assumes the domain of an input image is known. For simplicity, we only considered the scenario of incrementally learning *two* tasks; we randomly chose 2 out of 10 tasks in every run and reported the averaged results over 4 different runs. We denote the $t$-th task as $T_t$ with $t \in \{1, 2\}$.

**Baselines.** We adopt *fine-tuning* without any consideration of CL and three representative CL methods: LWF (Li and Hoiem 2017), EWC (Kirkpatrick et al. 2017), and ER (Chaudhry et al. 2019). LWF and EWC add regularization terms in their training objectives to penalize deviation from the past model and balance the stability-plasticity trade-off by controlling the regularization hyperparameter. On the other hand, ER stores some data from past tasks in a small exemplar memory and replays them while learning current task. For ER, we store 500 samples, which are 10% of a task data. Finally, as a model debiasing technique, we employ MFD (Jung et al. 2021), a state-of-the-art method that trains a domain-independent model using a MMD-based feature distillation method.

**Metrics.** We consider two metrics, average accuracy and the difference of classwise accuracy (DCA) (Berk et al.

2021), as evaluation metrics for CL performance and bias for each task, respectively. The concrete definition of DCA is given below. Let $\mathcal{D}_t = \{(x_t^{(i)}, a_t^{(i)}, y_t^{(i)})\}_{i=1}^{N_t}$ be a test dataset for task $T_t$, where $a_t^{(i)} \in \mathcal{A}$ is the color domain of the input $x_t^{(i)} \in \mathcal{X}$, and $y_t^{(i)} \in \mathcal{Y}_t$ is the class label where $\mathcal{Y}_t$ is the set of classes of $T_t$. Given a classifier $h$ and a dataset $\mathcal{D}_t$ for task $T_t$, DCA is defined as below:

$$\text{DCA}(h, \mathcal{D}_t) = \frac{1}{|\mathcal{Y}_t|} \sum_{y \in \mathcal{Y}_t} \max_{a,a' \in \mathcal{A}} |\text{A}(h, \mathcal{D}_t^{y,a}) - \text{A}(h, \mathcal{D}_t^{y,a'})|,$$

$$\text{where} \quad \text{A}(h, \mathcal{D}_t^{y,a}) = \frac{1}{N_t^{y,a}} \sum_{i=1}^{N_t^{y,a}} \mathbb{1}[h(x_t^{(i)}, t) = y]$$

in which $\mathcal{D}_t^{y,a}$ is the subset of $\mathcal{D}_t$ that is confined to the samples with class-domain label pair $(y, a)$. We note that $\text{A}(h, \mathcal{D}_t^{y,a})$ is the accuracy of samples with class $y$ and domain $a$, and DCA means the average (over class) of per-class maximum accuracy difference between domains. Thus, a large DCA corresponds to $h$ possessing large bias between different domains.

In addition, we compute forgetting ($\mathcal{F}$) and intransigence ($\mathcal{I}$) measures (Chaudhry et al. 2018; Cha et al. 2021) for evaluating stability and plasticity of a CL method, respectively, and use *Normalized* $\mathcal{F} - \mathcal{I}$ as a metric for the relative weight on plasticity and stability. To be specific, let $h_t$ and $h_t^*$ be the classifiers learned up to $T_t$ tasks which are trained by a CL method and the fine-tuning method, respectively. In our two task learning scenario, the forgetting and intransigence measures are defined as follows:

$$\mathcal{F} = \text{A}(h_1, \mathcal{D}_1) - \text{A}(h_2, \mathcal{D}_1) \tag{1}$$

$$\mathcal{I} = \text{A}(h_2^*, \mathcal{D}_2) - \text{A}(h_2, \mathcal{D}_2). \tag{2}$$

Then, for each CL scenario, the differences between two measures are normalized by the maximum and minimum possible values of $\mathcal{F} - \mathcal{I}$, which are obtained by $\text{A}(h_1, \mathcal{D}_1) - \text{A}(h_2^*, \mathcal{D}_2) + 1$ and $\text{A}(h_1, \mathcal{D}_1) - 1 - \text{A}(h_2^*, \mathcal{D}_2)$, respectively. We note that this Normalized $\mathcal{F} - \mathcal{I}$ indicates the model focuses more on stability as the value becomes lower and on plasticity as it becomes higher.

### Study 1: Forward Transfer of Bias

To investigate the influence of bias captured from $T_1$ in a CL scenario, we evaluated baseline methods by varying the bias level of $T_1$, while that of $T_2$ is fixed to level 2. Figure 1 shows DCA of $T_2$ along with Normalized $\mathcal{F} - \mathcal{I}$ after learning $T_2$ with two different bias levels of $T_1$, *i.e.*, level 0 & 6. The figure plots the results of LWF and EWC with various regularization strengths; namely, the upper the point is, the lower the regularization strength is. From the gap of blue triangles in the figure, we first observe that bias of $T_1$ adversely affects the bias of $T_2$, *i.e.*, forward transfer of bias exists, even with simple fine-tuning, which is consistent with (Salman et al. 2022). Second, we observe that when applying CL methods, the gaps between connected points get larger than fine-tuning. Moreover, when the bias level of $T_1$ is 6, DCA of $T_2$ for EWC and LWF increases more drastically as the focus

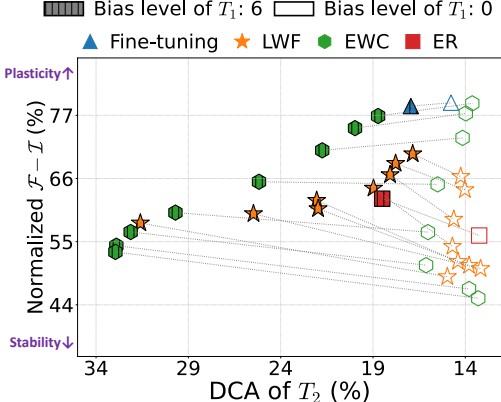

Figure 1: Forward transfer of bias. The higher DCA indicates a model is more biased. The y-axis shows the level of focus on plasticity or stability. Dashed lines connect the points with the same learning strategy (hyperparameters).

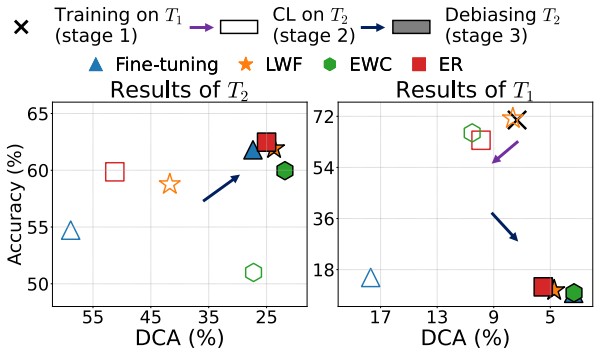

Figure 2: Backward transfer of bias. Blue arrows indicate the sequence of stages. Since all baselines are trained in the same way on $T_1$, we report the results with one cross marker.

on stability is larger. Thus, these results imply that CL methods promote the forward transfer of bias since they tend to remember the knowledge of past tasks for stability. Finally, we clearly observe that DCA of $T_2$ is always better when learned after $T_1$ with bias level 0 than with bias level 6, for similar Normalized $\mathcal{F} - \mathcal{I}$. Therefore, we argue that whenever a given task has a bias in CL scenario, its bias should be mitigated for learning future tasks.

**Study 2: Backward Transfer of Bias**

Here, we set the bias level of $T_1$ and $T_2$ as 0 and 6 and assume a scenario that a bias is detected after learning $T_1$ (stage 1) and continually learning $T_2$ (stage 2) by a CL method. In this situation, one may naively consider applying a debiasing method (stage 3), *e.g.,* MFD, to the model obtained after learning $T_2$ to remove the bias.

Figure 2 shows the accuracy and DCA of $T_1$ and $T_2$ on each stage for each baseline. In the right plot, we observe that points shift to the bottom left as they progress from stage 1 to stage 2. This means that as the stability gets less

focus, the bias obtained from $T_2$ is more transferred to $T_1$, *i.e.*, the backward transfer of bias occurs. Moreover, from the results of stage 3 in the left plot, we show that DCA of $T_2$ can be successfully reduced by employing a debiasing technique of a model with similar accuracy. However, we also identify that the accuracy of $T_1$ significantly drops after stage 3, which suggests that serious forgetting of $T_1$ happens when naively debiasing the model for $T_2$. This result suggests that just applying canonical debiasing of the model after learning each task, in order not to forward transfer the bias to the future tasks (as seen in Study 1), can cause serious forgetting of the past task. Hence, we argue that it is necessary to develop a novel continual learning method that takes causal learning into account to prevent the bias transfer while maintaining the stability of the model to mitigate forgetting.

**Concluding Remark**

With systematical analyses for two task CL scenarios using a synthetic dataset containing the color bias, we showed the bias can be transferred both forward and backward by typical CL methods that are oblivious to the dataset bias. Furthermore, we also showed that naively applying the existing debiasing technique inevitably leads to catastrophic forgetting, which strongly appeals for devising a new method that achieves objectives of CL and causal learning simultaneously. For future work, we will investigate the bias transfer in a more realistic scenario with natural biases such as gender bias or with a longer sequence of tasks. It would also be interesting research direction how the bias transfer matters when there is multiple sources of, possibly unknown, bias.

**Acknowledgments**

This work was supported in part by LG AI Research.

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
