# OpenReview forum: "Issues for Continual Learning in the Presence of Dataset Bias"
_AAAI.org/2023/Bridge/CCBridge — AAAI23 Bridge Continual Causality_

### Official Review · Reviewer_zMFD · 2022-11-28
**Please expand on the need for causal learning for CL+debiasing**

**Rating:** 6
**Confidence:** 3

**Review:**

The paper shows that CL methods inherently suffer from dataset bias, and that popular debiasing methods may help to mitigate the problem.

From a CL perspective, I find it interesting that bias transfers during CL. I would have expected this phenomenon intuitively but it’s nice to see it quantified.

The main issue that I have is that I don’t see a strong connection with causal learning. The paper argues that the results “strongly appeal the necessity of a novel method that can do causal learning while preventing forgetting”, but to me the results only say that we need to account for bias during CL training.
I would have liked a stronger focus on the link between debiasing and causal learning, and possibly novel problems that may arise from their combination.

Overall, I think the proposed direction (CL+causal learning for debiasing) may provide interesting discussion points for the bridge. If possible, I encourage the authors to expand more on possible research directions, synergies, and problems that may arise by combining CL+causal learning for debiasing.

---

### Official Review · Reviewer_b65R · 2022-12-01

**Rating:** 7
**Confidence:** 4

**Review:**

The paper studies the effect of the knowledge transfer when the training data is biased. In the paper, they show that knowledge learned from biased data can be forward and backward transferred.

Two well-designed experiments, where the bias is introduced by reducing the color discriminability of the image, show that biases can both be transferred in forwards and backward direction.

If space allows, it would be nice to include the equations for the metrics used.

I think the paper can lead to interesting discussion at the workshop.

---

### Decision · Program_Chairs · 2022-12-05

**Decision:**

Accept

**Comment:**

Accept - Poster

This paper discusses the effects of knowledge transfer when it is learned from biased data. The authors include well-designed experiments. The paper meets the overall themes of the bridge program and received positive reviews. We suggest that the authors use the additional space in the camera-ready version to integrate the reviewers’ comments and concerns, in particular expanding upon the connection with casual learning.